# Diet and Glycemic Index in Children with Type 1 Diabetes

**DOI:** 10.3390/nu15163507

**Published:** 2023-08-09

**Authors:** Alessia Quarta, Miriana Guarino, Roberta Tripodi, Cosimo Giannini, Francesco Chiarelli, Annalisa Blasetti

**Affiliations:** Department of Pediatrics, University of Chieti—Pescara, G. D’Annunzio, 66100 Chieti, Italy; alessiaquarta54@gmail.com (A.Q.); mirianag_@hotmail.it (M.G.); roberta.tripodi@studenti.unich.it (R.T.); cosimo.giannini@unich.it (C.G.); chiarelli@unich.it (F.C.)

**Keywords:** type 1 diabetes, diet, glycemic index, glycemic load, glycemic control, low-glycemic-index diet

## Abstract

In children with type 1 diabetes, a healthy lifestyle is important to control postprandial glycemia and to avoid hyperglycemic peaks that worsen the inflammatory state of vessels and tissues. Glycemic index and glycemic load are two important indexes which assess the quality and quantity of foods consumed during meals. The main macronutrients of the diet have a different effect on postprandial blood glucose levels, so it is important that diabetic children consume foods which determine a slower and steadier glycemic peak. In this review, we present the results of the most recent studies carried out in the pediatric population with T1D, whose aim was to analyze the effects of low-glycemic-index foods on glycemic control. The results are promising and demonstrate that diets promoting low-glycemic-index foods guarantee a greater glycemic stability with a reduction in postprandial hyperglycemic peaks. However, one of the main limitations is represented by the poor adherence of children to a healthy diet. In order to obtain satisfactory results, a possibility might be to ensure a balanced intake of low-, moderate- and high-glycemic-index foods, preferring those with a low glycemic index and limiting the consumption of the high- and moderate-glycemic-index types.

## 1. Introduction

Diabetes is a widespread metabolic condition all over the world [1,2]. In 2021, the new cases of children and adolescents with type 1 diabetes (T1D) were estimated to be about 355,900 globally, with the highest percentage of new cases in New Zealand, Western and Northern Europe and North America, and the lowest percentage in West Africa, South and East Asia and Melanesia, probably due to underdiagnosed cases [3,4]. This trend is still on the rise, with a prevision of 476,700 new cases in 2050 [5,6]. The optimal management of T1D includes both an appropriate pharmacological approach, through insulin regimens, and a balanced nutritional approach [7,8,9]. A lower adherence to recommendations was associated to a poorer glycemic control with postprandial hyperglycemic spikes, which contribute to endothelial damage and inflammation and an increase in oxidative stress that worsens the cardiovascular risk in patients with diabetes [10,11]. In this light, the proper control of postprandial blood glucose is important to reduce micro- and macrovascular damage, as evidenced in many studies in diabetic patients [12,13].

## 2. Diet in Children with T1D

Nutrition in children with T1D is a cornerstone in the management of the disease. Dietary recommendations for diabetic children are the same as for healthy children, and according to the “International Society for Pediatric and Adolescent Diabetes” (ISPAD) guidelines, the daily energy intake should consist of 45–50% of carbohydrates, 30–35% of fat (saturated fat <10%) and 15–20% of protein [14]. The main goal of a balanced diet is to contribute to an adequate glycemic control, ensure appropriate growth and protect against overweight and obesity. A proper dietary pattern involves eating three main meals and two snacks and having a regular meal-time routine. Breakfast represents the main meal and should provide about 30% of the daily caloric intake; dinner should be taken at least 2 h before bedtime [15,16,17,18,19]. Among macronutrients, carbohydrates are the main constituents of the diet constituting 45–50% of the daily energy intake and providing 4 Kcal/g [20]. Carbohydrates can be classified in several ways and the indexes expressing their qualitative and quantitative characteristics and effects on blood glucose are represented by the glycemic index (GI) and glycemic load (GL). Particular attention should be pointed at simple sugars, whose intake should not exceed 8–10% of the daily energy requirements, because their rapid absorption cause high and short glycemic peaks (Figure 1). High- and moderate-GI foods are not only represented by processed foods and sugary drinks, which are associated with weight gain and should be avoided; for example, some types of fruit cause glycemic spikes, while others, such as blueberries, raspberries and blackberries, result in a steadier rise in glycemia. It is essential to ensure an adequate daily intake of fruit (max 300 g/day) in order to provide a good supply of micronutrients such as vitamins and minerals. Finally, we emphasize the importance of dietary fibers, which determine a slower intestinal absorption of carbohydrates, resulting in a more stable glycemic peak [21]. Fats should account for 30–35% of daily caloric intake and provide 9 Kcal/g of energy. They are divided into saturated fats, found in animal products, and unsaturated fats, mostly contained in vegetable products. In a healthy diet, unsaturated fats, and particularly polyunsaturated fats (PUFAs), such as omega-3 and omega-6, should be preferred, as they are related to the maintenance of a good lipid balance and the reduction in cardiovascular risk. Moreover, fats are the source of the fat-soluble vitamins A, D, E and K and lead to a slower gastric emptying, helping to prevent glycemic spikes [22]. Finally, among the macronutrients, we mention proteins, which constitutes 15–20% of the daily intake and provides 4 Kcal/g of energy. Proteins are included in both animal (eggs, meat, fish) and vegetable foods, such as legumes. They mostly perform a building function by contributing to growth, although in some cases, such as in uncontrolled T1D, they are used as a metabolic source [9]. Regarding the effects on blood glucose levels, foods high in fat and/or protein have been shown to result in a delayed and prolonged rise in glycemia and this effect might probably be due to the action of fats to reduce gastric emptying time [20,23]. Consequently, preprandial insulin dosing should be adjusted by considering not only carbohydrates, but also other macronutrients in order to avoid hyperglycemic peaks; the possibility to perform a double bolus of ultra-rapid insulin with distanced administration should also be evaluated. Smart et al. analyzed the separate and combined effects on blood glucose of a high-fat and high-protein meal, in the presence of a constant carbohydrate count, in a group of 33 children aged 8–17 years with T1D on insulin therapy. It was observed that protein and fat had a delayed effect on postprandial glycemia: in particular, a high-protein meal rises glycemia 3 h later, having a sustained effect up to 5 h, and a high-fat meal 3.5 h later. The combined intake of high-protein and high-fat meals was associated with a greater increase in blood glucose in a range of 3 to 5 h [23,24]. Consequently, high-protein meals result in a prolonged glycemic rise up to 5 h, having protective effects for hypoglycemic episodes [25].

### 2.1. Carbohydrate Structure and Classification

Carbohydrates are the main macronutrients in diet, source of energy and regulators of glucose metabolism [26,27]. They constitute a large group of substances classified according to their chemical, physical and physiological properties [28,29]. Their chemical structure consists of carbon, hydrogen and oxygen atoms. Thus, they are differentiated into monosaccharides and disaccharides, also called simple sugars (consisting of 1–2 units), such as glucose, fructose, galactose, lactose, sucrose, etc. [30], oligosaccharides (short-chain carbohydrates consisting of 3–9 units) [31] and polysaccharides (long-chain carbohydrates consisting of >10 units). Polysaccharides can be classified into starch (amylose, amylopectin) and non-starch (dietary fibers such as cellulose). Starch is the main carbohydrate in diet and consists only of glucose molecules [32,33,34]. A more recent classification distinguishes carbohydrates as glycemic and non-glycemic. Carbohydrates that provide glucose molecules after digestion and absorption in the small intestine are called glycemic, whereas those that are not absorbed are called non-glycemic and are mainly represented by dietary fibers [35,36]. Most simple sugars, some oligosaccharides, such as maltodextrins, and starch can be classified as glycemic carbohydrates. The extent and manner in which carbohydrates rise glycemic levels compared to a reference carbohydrate (usually white bread or glucose) represent the GI of a food [37]. The glycemic response to a food containing carbohydrates depends on many factors such as the properties of the food and the metabolic characteristics of the individual. For example, simple sugars and starchy polysaccharides are more easily digested and, consequently, they are absorbed faster [38]. Gastric emptying time and intestinal transit time represent other major factors which regulate the patterns of blood glucose rise [39]. The GI would allow a better classification of carbohydrates according to their metabolic effects. Consequently, in recent years, this parameter is beginning to be taken into account in clinical practice, especially in the management of certain diseases such as diabetes.

### 2.2. GI and Glucose Blood Levels

According to the ISPAD, children with T1D must receive adequate energy and calorie intake to ensure optimal growth and they should maintain a good glycemic control [14,15,16,17,18,19,20,21,22,23,24,25,26,27,28,29,30,31,32,33,34,35,36,37,38,39,40]. We know that the quality and quantity of foods have a pivotal role in order to regulate the levels of postprandial blood glucose and daily insulin requirements in patients with diabetes [41]. Particularly, carbohydrates are the main macronutrient which influence glycemia after a meal [42]. It is well known that carbohydrates do not all have the same structure and properties; consequently, when counting carbohydrates, for the same grams, different glycemic responses may occur [43]. For the first time in 1981, Jenkins et al. developed the GI of a food, which represents the effect of a carbohydrate portion (usually 50 g) on postprandial blood glucose, compared with glucose or white bread [44,45]. GI depends on the value of area-under-the-blood-glucose curve (iAUC) of test food and reference food, expressed in percentage [46]:GI = (iAUC test food/iAUC reference food) × 100

The GL is the amount of carbohydrates that raises blood glucose postprandially [47]; in this view, the GI does not consider the effect of a specific amount of carbohydrates in a food portion on glycemia, but it is influenced by the timing that carbohydrates need to be broken down and absorbed in bloodstream [21]. GL is the result by the multiplication between GI and quantity of carbohydrates in a specific amount of food [46]:GL = GI/100 × quantity in grams of carbohydrates in a specific amount of food

The quality of carbohydrates in diet is important for glycemic control in patients with diabetes mellitus [48]. The fourth edition of the International Tables of Glycemic Index and Glycemic Load Values published in 2021 considers 4000 food items, more than that which was considered in a previous edition (2018) (Table 1) [49]. Generally, a high GI value is ≥70, a moderate GI value is between 56 and 69 and a low GI value is ≤55 [50]. The highest values of GI are among potatoes, rice and Indian–Asian regional food, while the lowest values are among meat products, dairy products and legumes. Bread, breakfast cereals and cereal grains have high GI values but there are differences between each food category [51]. Indeed, savory snacks have the highest GI value rather than sweet snacks and confectionery products [49]. This guide is important to establish a low-GI diet for patients with type 1 and 2 diabetes. Carbohydrates with a low GI determine a slower and more constant glycemic response, so it has been proposed that diets with low-GI foods could promote a better glycemic control in children with T1D [52,53] (Figure 2). Zafar et al. in their meta-analysis show that low-GI food improves metabolic profile in patients with diabetes, with a better insulin sensitivity, even if these modifications appear small [54]. However, there are discordant hypotheses, because one of the main limitations of this dietary approach would be related to a poor adherence by the pediatric population, resulting in an increased consumption of fatty foods. Furthermore, there are currently not many studies evaluating the long-term effect of low-GI diets in children with T1D.

## 3. Diet and Growth in Children with T1D

Linear growth in children is a complex developmental process regulated by genetic, endocrinological and nutritional factors [55]. In children with T1D, achieving an appropriate height target represents a major challenge in the management of the disease. Indeed, it is well known that inappropriate glycemic control associated with a state of chronic hyperglycemia impairs linear growth and pubertal development. Linear growth is mainly regulated by the growth hormone (GH)/insulin growth factor-1 (IGF-1) axis, which acts on the epiphyseal growth plate. GH is secreted by the somatotropic cells of the adenohypophysis and regulates hepatic synthesis of IGF-1, which circulates bound o respective binding-proteins, like IGFBP-3. Insulin modulates the GH/IGF-1 axis through several mechanisms: it ensures an adequate serum concentration of IGF-1 and IGFBP-3 and increases the expression of peripheral GH receptors. In diabetic patients with poor metabolic control, an increased inflammatory state, lower levels of IGF-1 and reduced peripheral sensitivity to GH, resulting in GH hypersecretion and insulin resistance, have been observed. This condition is exacerbated during the pubertal phase, when higher GH peaks occur with an increased risk of hyperglycemia and poor metabolic control if insulin regimes are not adequate [56,57]. In recent years, advances in blood glucose monitoring techniques and insulin treatment have enabled the achievement of better glycemic control with a better prognosis of growth. Other factors associated with poor linear growth in children with T1D are improper eating habits. Indeed, a proper diet in children with diabetes should include a balanced intake of macronutrients and micronutrients. In recent years, various dietary patterns have been proposed that could improve glycemic control in patients with T1D. Low- and very-low-carbohydrate diets have been associated with a greater glycemic stability and better HbA1c values. However, despite the positive effects, they are not recommended in children with T1D, as they would not ensure the adequate daily caloric intake with an increased risk of growth failure [58]. Conversely, balanced diets with a preference for low-GI foods ensure an appropriate nutrient intake and greater glycemic control by avoiding postprandial hyperglycemic states, promoting the physiological functioning of the GH/IGF-1 axis and an adequate growth [59].

## 4. Studies Focusing on Diets Characterized by Different Quantities and Qualities of Carbohydrates in Children with T1D

One of the main therapeutic objectives in patients with T1D is to obtain a stable glycemic control in order to reduce the risk of the onset of long-term micro- and macrovascular complications [60,61,62]. Stable glycemic control can be achieved via a balance between pharmacological approach and nutrition [63,64]. However, despite medical advances, only 20% of children with T1D succeeds in reaching a target of HbA1c <7.5% [65,66]. Much interest has been given to diets characterized by a reduced carbohydrate intake, because it has been observed that lower carbohydrate consumption is associated with a greater glycemic stability [67]. Many studies have been conducted, which analyzed the effect of different types of dietary approaches in children with diabetes (Table 2).

### 4.1. Studies Analyzing the Effects of Carbohydrate-Restricted Diets

Carbohydrate-restricted diets are represented by low-carbohydrate diet (LCD) and very-low-carbohydrate diet (VLCD). LCD usually provides a daily consumption of 30% of carbs (about 100 g/day), 50–60% of fat, and 20–30% of protein. VLCD is characterized by a daily consumption of carbs of about 50 g/day and is therefore considered a ketogenic diet. Recent studies have analyzed the benefits and risks of these types of diet, especially in children with T1D [68]. Lennerz et al. conducted an online survey to evaluate the effect of a VLCD in pediatric and adult patients with T1D. The participants followed a diet with a daily consumption of 36 ± 15 g of carbohydrates/day for a period of 2.2 ± 3.9 years. An excellent effect on glycemic control and postprandial glycemia has been observed, as well as an HbA1c of 5.67 ± 0.66% and a 1% incidence of diabetic ketoacidosis (DKA) and hypoglycemia. Moreover, a reduced daily insulin requirement and increased insulin sensitivity have been observed. Regarding lipid profile, normal triglyceride values and high HDL and LDL cholesterol levels have been observed [69]. However, the long-term effects and safety of a carbohydrate-restricted diet, especially in the pediatric population, are still unknown. Indeed, this type of diet is discouraged in children due to the potential risk of DKA, dyslipidemia, inappropriate caloric intake, adverse effects on growth and pubertal development and a patient’s poor diet adherence [70]. Lejk et al. also evaluated the impact of specific LCD (30% of daily caloric intake compared to 50%) on glycemic control in a small population of Polish children with T1D and continuous blood glucose monitoring (CGM). According to the Clinical Recommendations of Diabetes Poland 2021, carbohydrates should represent 45–50% of the daily energy intake and the amounts of simple sugars should not exceed 10%. A better glycemic control with a higher time in range (TIR) and better values of HbA1c were observed in the group following LCD. It was also highlighted that eating balanced meals, with a predilection for carbohydrates with a low GI, would result in fewer glycemic fluctuations and a lower risk of hypo- and hyperglycemia [71]. Cherubini et al. evaluated the association between macronutrient intake and TIR in a sample of Italian children and adolescents with T1D and CGM. Although all macronutrients affect metabolic control, carbohydrates are the main determinants of postprandial glycemia. According to the most recent ISPAD 2018 guidelines, the recommended proportion of carbohydrates should be 45–50% of the daily caloric intake. In this study, a higher TIR and target HbA1c were observed in patients with a daily carbohydrate intake of 40–44% compared to 45–50%. Similarly, a diet higher in fat and too low in carbohydrates would appear to be associated with a worse glycemic control. However, these data should be implemented with studies including other populations with different socioeconomic and ethnic backgrounds [72]. Neuman et colleagues, in their multicenter study, investigated the effects of LCD through a questionnaire addressed to a caregiver of 1040 children with T1D. Compared with children following a normal diet, it was found that the LCD group had comparable HbA1c values, a higher TIR and fewer episodes of hyperglycemia, but greater episodes of hypoglycemia. Therefore, although associated with a modestly improved blood glucose control, LCD should be used with caution in children because of the increased risk of hypoglycemia [73]. We can conclude that, although carbohydrate restriction diets are associated with a better glycemic control, they are not recommended in the pediatric population. Indeed, these diets are associated with an increased risk of onset of complications such as hypoglycemia, DKA, dyslipidemia and a growth failure due to nutritional deficiencies, but also poor adherence due to reduced food choices. In addition, the adoption of this type of diet might also have behavioral repercussions by causing increased stress levels, which in turn would result in a worse glycemic control [74,75,76,77].

### 4.2. Studies Analyzing the Effects of Low-GI-Index Diets

Although the importance of nutrition in the multicenter management of diabetes is well known, the association between the qualitative characteristics of the diet and the effect on glycemic control is still controversial. Many studies have been conducted, which focused on the effects of diets characterized by the consumption of low-GI foods. In their study, Rovner et al. compared the effect of a diet with low-GI foods versus a normal diet in a small cohort of children with T1D and CGM. It was observed that the consumption of low-GI foods compared to the usual diet was associated with lower postprandial glycemic values and less glycemic variability. Children who followed this diet consumed more complex carbohydrates, such as fibers, and less fat, but the overall intake of calories, carbohydrates and proteins did not vary. Once again, despite the promising results, longitudinal studies would be needed to verify the long-term metabolic effects and children’s adherence to this type of diet [78]. Marquard et al. compared the effects of a low-GI diet with an ISPAD-recommended optimized mixed diet (OMD) in children with T1D. It is well known that a low-GI diet reduces blood glucose fluctuations and improves the glycemic profile compared to a high-GI diet; however, the benefit of a low-GI diet compared to OMD is controversial. In this study, it was shown that there were no significant differences between the two groups regarding HbA1c levels, but in the low-GI diet group, carbohydrate intake was reduced and fat intake increased, and in the OMD group, energy and fat intake were both reduced. Consequently, the macro- and micronutrient composition of an OMD diet was shown to be better than that of a low-GI diet for children with diabetes. Moreover, an OMD diet would have fewer restrictions than the low-GI diet, allowing for a better adherence and easier home management of the condition [79]. According to the recommendations of the American Diabetes Association (ADA), more importance should be given to the quantity rather than the quality of carbohydrate intake for the therapeutic management of T1D [80,81]. However, there are many studies in the literature which aim to demonstrate the positive effects of low-GI diets, although it remains controversial whether this type of diet may contribute to glycemic control together with insulin-to-carbohydrate dosing therapy, and if the GI may influence insulin requirement. A comparison between low-GI and high-GI diets was performed by Wang et al. In their meta-analysis including nineteen studies, they analyzed the effects of the two types of diets on HbA1c and fructosamine levels in patients with T1D and T2D. It was observed that HbA1c and fructosamine were lower in patients who followed a diet with low GI. A low-GI diet would have benefits both for glycemic control and the management of overweight/obesity in children [82]. Among studies conducted on patients with T1D, we mention Nansel et al. They analyzed the effects of low- and high-GI diets in a group of 20 young people with T1D, CGM and following the basal–bolus insulin regimen. It was observed that the low-GI diet resulted in lower glycemic levels during the day and reduced daily glycemic fluctuations without an increased risk of severe hypoglycemia. However, it was evidenced that it increased the frequency of moderate hypoglycemia with the necessity to reduce the dosage of rapid insulin. Therefore, a low-GI diet allows us to reduce the daily insulin requirement and to have a better glycemic control [83]. In 2016, Nansel et al. conducted a randomized clinical trial in a population of 136 children to evaluate the effect of nutritional interventions on glycemic control. It was seen that both diet quality and macronutrient quantity were associated with an effect on glycemic control; particularly, a diet with an increased intake of carbohydrates, plant fibers and fruits and that was low in fat and simple sugars would appear to be associated with a reduction in HbA1c [84]. Gilbertson et al. evaluated the long-term effects on the metabolic control and quality of life of a flexible low-GI diet versus a traditional carbohydrate-counting diet in children with T1D. A total of 104 children aged 8–13 years were recruited for a prospective randomized study in which the effects of the two types of diets on HbA1c values, insulin requirements, glycemic fluctuations and quality of life were analyzed over a 12-month period. It was observed that children following a flexible low-GI diet had a significant improvement in HbA1c values and a reduction in hyperglycemic episodes, while there were no differences in insulin requirements and hypoglycemic episodes. One of the main limitations of a low-GI diet in the pediatric population is that it may restrict food choice options and increase the consumption of fatty foods; however, in this study, it was observed that a more flexible dietary regimen with a preference of low-GI foods would be associated with a better perceived quality of life for children and parents [85,86]. In their study, Weyman-Daum et al. showed the effects of a low- and high-GI breakfast on postprandial glycemia in a group of 22 children with T1D not well controlled. Patients who received a low-GI breakfast ate an apple and bran cereal, whereas the second group had cornflakes and a banana. It was observed that children who ate the high-GI breakfast presented a higher glycemic peak than those who ate the low-GI meal; however, these differences did not occur when adjusted doses of preprandial insulin were administered compared to the usual non-adjusted dose. Therefore, it was concluded that postprandial glycemia is most influenced by adequate doses of preprandial insulin administered rather than the GI of foods. However, the data obtained from this study relate to a short time span and cannot be taken into account for the evaluation of long-term effects [87]. Ryan et al. in their study tested the effect of the GI on postprandial glycemic variations in children with T1D and CGM, receiving multiple daily injections and determined the optimal insulin therapy for a low-GI meal. The study population consisted of 20 children who were offered for four consecutive days a meal with the same amounts of macronutrients but a different GI preceded by the administration of preprandial ultra-short-acting insulin; the low-GI meal was also consumed with preprandial regular insulin and postprandial ultra-short-acting insulin. It was observed that children who consumed the low-GI meal presented fewer glycemic fluctuations and postprandial glycemic peaks than the high-GI meal group. The most appropriate insulin regimen for a low-GI diet has been seen to be preprandial ultra-short-acting insulin [88]. In contrast to carbohydrate-restricted diets, we can state from the analyzed results that low-GI diets provide a benefit for glycemic control being also associated with lower and less severe risks. A limitation might be represented by children’s poor adherence to diet. Consequently, this type of diet could be recommended for children with T1D, although the necessity for further longitudinal studies examining the long-term metabolic effects remains.

**Table 2 nutrients-15-03507-t002:** Summary of the main studies on the effects of carbohydrate-restricted and low-GI diets on glycemic control in children with T1D. Abbr. T1D: type 1 diabetes; DKA: diabetic ketoacidosis; CGM: continuous glucose monitoring; GI: glycemic index.

Population	Type of Diet	Method	Results	Limitations	Ref.
**Children and adults with T1D**	Very-low-carbohydrates diet (VLCD)	Evaluation of HbA1c, insulin requirements and adverse events with VLCD diet through an online survey	Good glycemic control of T1D with low rates of adverse events	Potential risk of DKA, dyslipidemia, inappropriate caloric intake, adverse effects on growth and pubertal development and patient’s poor diet adherence	[69,70]
**Children with T1D and CGM**	Low-carbohydrates diet	Impact of low-carbohydrate diet on glycemic control	Higher time in range (TIR) and better values of HbA1c	Patient’s poor diet adherence	[71]
**Children with T1D and CGM**	Low-GI foods	Effects of low-GI foods compared to the usual diet	Lower postprandial glycemic values and less glycemic variability	Not known long-term metabolic effects and poor children adherence	[78]
**Children with T1D**	Low-GI diet	Effects of low-GI foods compared to ISPAD-recommended optimized mixed diet (OMD)	No significant differences about HbA1c levels; fat intake increased in the low-GI diet group and energy and fat intake reduced in OMD group	Patient’s poor diet adherence	[79]
**Young people with T1D, CGM and following the basal–bolus insulin regimen**	Low-GI diet	Comparison between low- and high-GI diets	Lower glycemic values, daily glycemic fluctuations, no increased risk of severe hypoglycemia	-	[83]
**Children with T1D**	Flexible low-GI diet	Evaluation of long-term effects (12 months) on metabolic control and quality of life of a flexible low-GI diet versus a traditional carbohydrate-counting diet	Significant improvement in HbA1c values and reduction in hyperglycemic episodes; no differences in insulin requirements and hypoglycemic episodes	Restricted food choice options and increased consumption of fatty foods	[85,86]
**Children with T1D, not well controlled**	High- and low-GI breakfast	Effect of a low- and high-GI breakfast on postprandial glycemia	Postprandial glycemia resulted most influenced by adequate doses of preprandial insulin administered rather than the GI of foods	-	[87]
**Children with T1D and CGM on basal–bolus insulin regimen**	High- and low-GI breakfast	Effect of a meal with the same amounts of macronutrients and different GIs	Fewer glycemic fluctuations and postprandial glycemic peaks in low-GI meal group	-	[88]

## 5. Conclusions

T1D is an autoimmune disorder characterized by an impairment of glucose metabolism due to a progressive loss of beta-pancreatic cells with consequent insufficient insulin secretion. The main challenge in the management of T1D in children is to achieve a stable glycemic control in order to ensure adequate growth and pubertal development and to avoid the onset of long-term micro- and macrovascular complications later in life. The main cornerstones for the management of T1D are insulin therapy and nutrition. A novelty is that, in recent years, the concept of GI has become more relevant by also focusing on the quality of carbohydrate intake. Many studies have been conducted on children with T1D, analyzing the possible effect of different types of diet on glycemic control. The results obtained are promising: data showed a significant improvement in glycemic control with a reduction in glycemic fluctuations and postprandial glycemic peaks after the intake of low-GI diets. In addition, a reduction in mean HbA1c values, reduced insulin requirements and improved insulin sensitivity were also observed, probably also due to the beneficial effect on overweight and obesity. The main limitation remains the child’s poor adherence to the diet due to restrictions in food choices, leading to a potential increased intake of fatty foods. A possible solution would be to not completely ban high- and moderate-GI food intake, but to limit it, achieving a balanced intake of high- and low-GI foods. It is also important not to reduce the amount of carbohydrates in the normal daily caloric intake, as children must always be ensured an adequate energy intake for optimal growth and development. Finally, it could be concluded that a diet characterized by a balanced intake of all macronutrients, promoting foods containing low-GI carbohydrates and limiting the high-GI ones, could be an important adjuvant together with insulin therapy for achieving glycemic balance in children with T1D.

## Figures and Tables

**Figure 1 nutrients-15-03507-f001:**
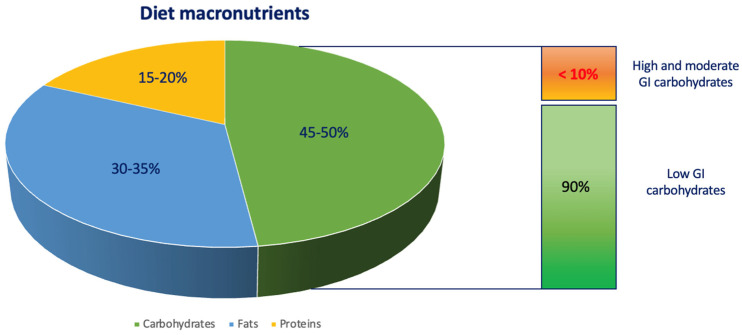
Distribution of macronutrients according to total daily energy intake.

**Figure 2 nutrients-15-03507-f002:**
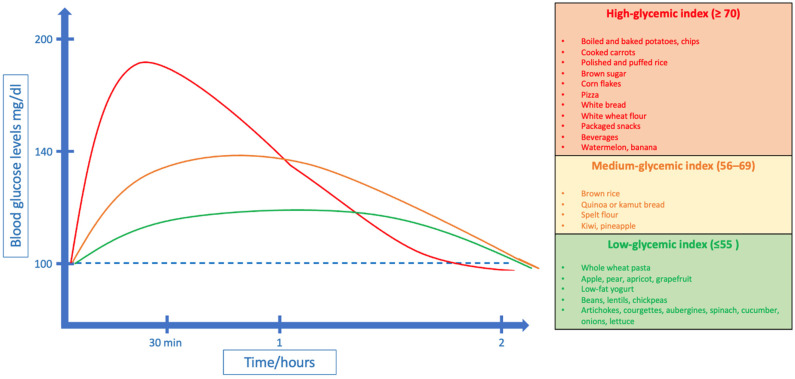
Glycemic peak after ingestion of high-, medium- and low-GI foods.

**Table 1 nutrients-15-03507-t001:** Mean of glycemic index of the main food categories.

Food Category	Mean GI
Bakery products	58
Beverages	50
Breads	64
Breakfast cereals	61
Rice	67
Cookies	49
Crackers	55
Dairy products	35
Fruits	51
Legumes	34
Nuts	22
Pasta	52
Savonary snacks	60
Snack bars	44
Soups	49
Potatoes	71
Other vegetables	66

## Data Availability

All data are contained within the article.

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
