# Peer review of "Diet and Glycemic Index in Children with Type 1 Diabetes"

_nutrients, 2023, doi:10.3390/nu15163507_

Round 1

Reviewer 1 Report

The authors have addressed a very important topic in their review paper focusing on a balanced diet and its importance in glycemic control in children with type 1 diabetes. 

The outline of the manuscript is satisfactory, it provides extensive and understandable insight on the impact of food with low glycemic index in DMT1 children. 

I would only suggest giving clear recommendations on what balance diet includes, and which quantity of high and moderate glycemic index carbs is allowed per day compared to low glycemic carbs. In addition, figure or table with those recommendations would improve quality of the paper. 

English language requires moderate editing, in some parts it seems crude. 

Author Response

Thank you for your suggestions, they were very helpful in improving and clarifying our manuscript.

The 2018 ISPAD guidelines contain the main recommendations for having a balanced diet, but they must be placed in a more complex context in which numerous other factors need to be considered, such as individual preferences and ethnic and socio-cultural background. The result is an individualized diet with the goal of ensuring good glycemic and metabolic control, proper growth and psychological health.

Unfortunately, there are no clear recommendations or specific guidelines on the daily amount of high and moderate GI carbohydrates allowed per day. However, if we consider sucrose as the reference food of high and moderate GI carbohydrates, the 2018 ISPAD guidelines recommend not to exceed 10% of the daily energy intake of sucrose (disaccharide with GI = 68±5). However, among high/moderate GI foods, a distinction must be made between processed industrial foods and sugary drinks, which are associated with weight gain and should be avoided, and other foods such as certain fruits (pineapple, bananas, cantaloupe, watermelon), grains, and vegetables (cooked carrots, potatoes), whose intake is still recommended.

Therefore, a diet is proper when a balanced association of the various macronutrients is ensured in every main meal. Indeed, the association of high/moderate GI, low GI and fiber-rich carbohydrates allows modulation of postprandial glycemic peaks because fibers slow intestinal absorption, just as fats slow gastric emptying.

These considerations are discussed in the section entitled "Diet in children with T1D."

Besides, we added a figure describing the daily quantitative distribution of macronutrients, also referring to the suggested daily amounts of high/moderate and low glycemic index carbohydrates.

Finally, we made a revision of the English language, trying to make the language of the manuscript more fluent.

Reviewer 2 Report

In the review article "Diet and glycemic index in children with type 1 diabetes", Alessia et al., comprehensively review different diets in diabetic children. Importantly, they review diet and growth in children with T1D. Along with an overview of studies focusing on diets characterized by different quantity and quality of carbohydrates in children with T1D, the authors include a section summarizing the effects of carbohydrate-restricted and low-GI diets. Overall, I believe this review will add greatly to the field of diets and diabetic children.

Author Response

Thank you for your positive comments. We are really glad that our manuscript was appreciated and considered complete and exhaustive. The achievement of a good glycemic control in children with T1D is still a great challenge. We sincerely hope that the topics discussed in this review and the suggestions provided would might be helpful in reaching this objective.